Fluorescent protein tagging confirms the presence of ribosomal proteins at Drosophila polytene chromosomes

Rugjee Kushal Nivriti
Roy Chaudhury Subhendu
Al-Jubran Khalid
Ramanathan Preethi
Matina Tina
Wen Jikai
Brogna Saverio s.brogna@bham.ac.uk
University of Birmingham , School of Biosciences , UK
Gregan Juro
Electronic publication date: 2013 Feb 12
Publication date: 2013
Volume: 1
Electronic Location ID: e15
Received 2012 Dec 3; Accepted 2013 Jan 3
Copyright: © 2013 Rugjee et al.
Copyright year: 2013
Copyright holder: Rugjee et al.
License: This is an open access article distributed under the terms of the Creative Commons Attribution License, which permits unrestricted use, distribution, and reproduction in any medium, provided the original author and source are credited.
License URL: https://creativecommons.org/licenses/by/3.0/

Keywords: Ribosomal proteins, Drosophila, Polytene chromosomes, Visualization

Funding: Royal Society fellowship and a Wellcome Trust This work was supported by a Royal Society fellowship and a Wellcome Trust grant to S.B that covered J.W. K.A. was supported by a Saudi Embassy PhD studentship. The funders had no role in study design, data collection and analysis, decision to publish, or preparation of the manuscript.

==============================
Most ribosomal proteins (RPs) are stoichiometrically incorporated into ribosomal subunits and play essential roles in ribosome biogenesis and function. However, a number of RPs appear to have non-ribosomal functions, which involve direct association with pre-mRNA and transcription factors at transcription sites. The consensus is that the RPs found at these sites are off ribosomal subunits, but observation that different RPs are usually found together suggests that ribosomal or ribosomal-like subunits might be present. Notably, it has previously been reported that antibodies against 20 different RPs stain the same Pol II transcription sites in Drosophila polytene chromosomes. Some concerns, however, were raised about the specificity of the antibodies. To investigate further whether RPs are present at transcription sites in Drosophila, we have generated several transgenic flies expressing RPs (RpS2, RpS5a, RpS9, RpS11, RpS13, RpS18, RpL8, RpL11, RpL32, and RpL36) tagged with either green or red fluorescent protein. Imaging of salivary gland cells showed that these proteins are, as expected, abundant in the cytoplasm as well as in the nucleolus. However, these RPs are also apparent in the nucleus in the region occupied by the chromosomes. Indeed, polytene chromosome immunostaining of a representative subset of tagged RPs confirms the association with transcribed loci. Furthermore, characterization of a strain expressing RpL41 functionally tagged at its native genomic locus with YFP, also showed apparent nuclear accumulation and chromosomal association, suggesting that such a nuclear localization pattern might be a shared feature of RPs and is biologically important. We anticipate that the transgenes described here should provide a useful research tool to visualize ribosomal subunits in Drosophila tissues and to study the non-ribosomal functions of RPs.

Introduction

Ribosomal proteins (RPs) are essential components of the ribosome – the universally conserved machine that translates gene information into proteins. Interestingly, in all organisms studied, it has been recently reported that many RPs also possess extra-ribosomal functions. In particular, a number of RPs regulate their own expression by binding their mRNA, pre-mRNA or promoter (Warner & McIntosh, 2009). In mammalian cells there is also evidence of RPs that regulate other genes by binding directly to specific transcription factors. RpL11, for instance, associates with a defined domain of the oncoprotein cMyc and inhibits transcription activation of cMyc target genes (Dai et al., 2007; Dai, Sun & Lu, 2010). Another example is RpS13, which in mammalian cells, binds its own pre-mRNA and inhibits splicing (Malygin et al., 2007). There is abundant evidence that these additional functions of RPs are unrelated to their role in ribosomal translation and are specific to individual RPs. For example the effect of RpL11 on cMyc is specific for RpL11 since other RPs do not show similar activities. There is also evidence that some RPs might bind chromatin-associated proteins in Drosophila and regulate transcription (Ni et al., 2006; Coleno-Costes et al., 2012). Tri-methylated RpL12, for example, interacts with the Corto protein at the chromosomes and regulates a set of heat response and RP genes (Coleno-Costes et al., 2012).

The current understanding is that free RP molecules not assembled into ribosomal subunits mediate extra ribosomal functions of RPs at chromosomes. Another key assumption is that only specific proteins associate at the particular gene loci, such as RpL11 at cMyc target genes. Although RPs usually co-purify along with chromatin, transcription factors and pre-mRNA processing factors, their presence is usually regarded as a contamination of these abundant proteins (Gavin et al., 2002; Jurica & Moore, 2003; Shi et al., 2009). This conclusion is further corroborated by the fact that the whole complement of 40S and 60S RPs is not observed. Yet it is plausible that complement sets containing RPs were actually recruited to the transcription sites but eventually dissociated along the course of the lysate extraction, leaving only those tightly-binding RPs behind at those sites (De & Brogna, 2010). For example, even though RpL11 has been consistently observed as the sole RP interacting with cMyc, it can be hypothesised that it was initially recruited at target genes as part of a complex, which then dissociated during the cell lysate extraction (De & Brogna, 2010). On the ribosome, RpL11 is found associated with the 5S rRNA, which together with other RPs are known to disassociate in the presence of EDTA (Steitz et al., 1988); indeed, EDTA was present in the lysis solution used by Dai, Sun & Lu (2010). The same logic can be applied for the sole presence of RpS13 observed at splicing sites, that is, RpS13 was initially recruited there as part of the 40S subunit, but except for RpS13, everything else was lost during the course of the experimental preparation (De et al., 2011).

Supporting this alternative explanation, a recent genome-wide ChIP-on-chip study provides evidence that RpL11 and two other 60S RPs tend to associate with the same sub-set of specific chromosomal loci (De et al., 2011). The study suggests that these three proteins are recruited to chromosomes as RP complexes. Furthermore, it has been previously reported that 21 RPs and rRNA have been observed at several sites on the polytene chromosomes, whereby RNA sensitivity and recruitment rates have indicated that their interaction is with nascent mRNAs. The combined presence of both RPs and rRNA at these sites argue for the presence of ribosomal-like subunits. However, there have been criticisms that the antibodies raised against the individual RPs may not be sufficiently specific and these would therefore cross-react with unspecific epitopes around the nascent mRNAs (Bohnsack et al., 2002). Here, to further study the association of ribosomal proteins with chromosomal loci in Drosophila, we have generated a number of transgenic flies carrying UAS-driven constructs expressing RPs tagged with either GFP or RFP, and characterized their expression in salivary gland cells. Additionally, we have analyzed the expression of RpL41 which was tagged at its native genomic locus,

Results

Ribosomal proteins tagging

Although most ribosomal proteins are essential for viability, several studies have indicated that it is feasible to tag the termini of a number of these proteins with GFP or other peptides without preventing sufficient incorporation into functional ribosomes (Hurt et al., 1999; Inada et al., 2002; Lam et al., 2007). In a similar fashion, in an attempt to develop tools that allow ribosome visualization in Drosophila cells, we tagged with fluorescent proteins several RPs that localize on either of the 40S or 60S subunits (Fig. 1A shows the positions of the RPs on the 80S; Fig. 1B shows a list of these proteins and their differing nomenclature across three model systems). Initially, we generated constructs expressing RpS9, RpS15, RpS18 and RpL11 tagged at either the carboxy or amino terminal with Green Fluorescent Protein (GFP) (Fig. 2A). To test their functionality, the constructs were first transfected into S2 cells and the expression assayed by Western blot analysis using an antibody against GFP (Fig. 2B). These four constructs produced bands of the right sizes, confirming that they are well expressed in S2 cells. Visualization of the GFP signal revealed most cells had the expected sub-cellular localization pattern: most of the signal was in the nucleolus and in the cytoplasm (Figs. 2C and 2D). The pattern of sub-cellular signal of the tagged RPs, suggested that the proteins might be functional. The observation that GFP, when not fused to any RPs, accumulates all over the cytoplasm and nucleus but without the characteristic nucleolar enrichment (Fig. 2C, top panels) also argues that the tagged RPs must retain the ability to bind rRNA. Notably, the GFP-tagged ribosomal proteins were also detected in the DAPI-stained region of the nucleus (Figs. 2C and 2D). The level of the nuclear fluorescence varied between cells and this depended on the transfection conditions: more efficient transfection reagents resulted in an increase in the number of cells with high fluorescence throughout the nucleus without the characteristic nucleolar enrichment (data not shown). This observation argues that because of over-expression, a large fraction of the tagged proteins must have failed to be incorporated into ribosomes and associated non-specifically with chromatin (see Discussion).

Figure 1 Location on the 80S and nomenclature of selected RPs.

(A) Front and back view diagrams showing the locations of some of the RPs on the eukaryotic 80S ribosome. The structures were generated with PyMol, by modifying a PyMol Session downloaded from http://www.mol.biol.ethz.ch/groups/ban_group/Ribosome, which visualizes PDB files 2XZM (40S) and PDB 4A17, 4A19 (60S). (B) Table with leftmost column listing the Drosophila RP studied in this paper; alternative names are provided within square brackets (information derived from Flybase). In the middle and rightmost column are listed orthologous RPs in Saccharomyces cerevisiae and Escherichia coli respectively (if present). The orthologies were confirmed by BLAST sequence search and alignment.

Figure 2 Expression of GFP-tagged RPs in S2 cells.

(A) Schematic of the constructs expressing the indicated GFP-fusion RPs, either tagged at the C- (top) or N-terminal (lower), both under the control of the Drosophila Actin 5C promoter (Ac5). All constructs included the late SV-40 polyadenylation signal at the 3’ end. (B) Western blot analysis of lysates from transiently transfected S2 cells expressing the proteins indicated above the lanes. The lane labelled B46 (GFP) is an extract from cells expressing untagged GFP. Similar amount of extracts were loaded in all lanes. The proteins were detected with an anti-GFP antibody. (C) The leftmost panels show micrographs of cells expressing C-terminally tagged RPs. Middle, DAPI stain showing the nucleus. Rightmost, are shown merged images of the GFP and DAPI signals. The top row shows cells expressing GFP alone. (D) The leftmost column shows cells expressing N-terminally tagged RPs.

To further assess the functionality we analyzed the association of the tagged RPs with ribosomes. To achieve this, cell extracts were separated by sucrose gradient centrifugation and the fractions – corresponding to free proteins, 40S, 60S, 80S and polysomes were analysed by Western blotting with a GFP specific antibody (Fig. 3 and Material and Methods). We found that a proportion of all four tagged RPs co-migrated with polysomes, monosomes and ribosomal subunits fractions as expected (Fig. 3); however, for RpS9 and RpL11 a substantial fraction of the protein is found in lighter non-ribosomal fractions and for all proteins there was a low correlation between the UV trace and protein level. It seems that in these transfected cells only a fraction of the tagged RPs gets incorporated into the corresponding ribosomal subunits, while the remaining either accumulate as free proteins in the nucleus or form large non-ribosomal complexes that co-migrate with polysomes. Therefore, whereas this analysis suggests that using transient transfection can lead, probably because of over expression, to the mislocalization of the tagged RPs, the partial association with ribosomes and the characteristic cytoplasmic/nucleolar sub-cellular localization pattern seen in most of the cells, argues that under more optimal expression levels the tagged proteins should remain sufficiently functional and can compete with the endogenous RPs and assemble into ribosomes.

Figure 3 GFP-tagged RPs are present in ribosomal fractions.

Polysome analysis of extracts from cells transiently transfected with constructs expressing the RPs indicated above each graph. Extracts were separated through a 50%–10% sucrose gradient and fractionated while recording OD254 reading (traces). Fractions (1 mL) were precipitated and analyzed by Western blot using anti-GFP (signal shown below each trace).

Generation and analysis of transgenic flies expressing GFP or RFP tagged ribosomal proteins

To study RPs in a more physiological and amenable experimental system than transiently transfected cells, we have generated transgenic flies expressing tagged versions of the RPs described above as well as several more small and large subunit proteins (listed in Fig. 1B). Small subunit proteins were tagged with GFP while those in the large subunit with RFP. The transgenes were generated by germ line transformation, using either P-element mediated integration or the site-specific PhiC31 system (Materials and Methods). In all instances, expression of the constructs was under the control of the regulatable UAS promoter. The GAL4/UAS system allowed expression in the salivary glands by crossing the UAS transgenes with a strain expressing Gal4 in salivary glands throughout larval development (Material and Methods). The salivary glands are made of very large cells with polytenic nuclei and therefore make an amenable system to visualize proteins in the nucleus (see below).

The transgenes encoding RpL11-RFP, RpS18-GFP and RpS9-GFP were first crossed with a strain carrying GMR-Gal4, which is expressed at a high level in the developing eye (Freeman, 1996). Over-expression in the eye is a very sensitive assay to visualize eventually toxic effects associated with the expression of transgenes (Freeman, 1996). None of the transgenes showed roughening of the eye, indicating that over-expressing these fusion proteins is not detrimental for the cell (data not shown). Next, the transgenic flies were crossed with a salivary gland specific driver strain (SG-Gal4, Material and Methods) which allows tissue-specific expression in the salivary glands from mid third-instar larvae. Fluorescence imaging showed a sub-cellular localization pattern of small subunit RPs (RpS9, RpS18, RpS13, RpS2, RpS11, and RpS5a) fused with GFP (Fig. 4) and large subunit RPs (RpL36, RpL11, RpL8, and RpL32) fused with RFP (Fig. 5). Unlike in transfected S2 cells where not all cells show a high concentration of RPs in the nucleolus, in salivary gland calls, all RPs show a prominent signal in the nucleolus where most of the events of ribosome biogenesis take place. The signal in the nucleolus was typically stronger than that in the cytoplasm. The seemingly dimmer cytoplasmic signal, though, is probably a visual effect from the fact that the salivary glands cells are replete with vesicles. There is an apparent negative correlation between the number and size of the vesicles, and apparent cytoplasm signal (for example, compare the RpS2 and S11 micrographs in Fig. 4). At this stage (third instar), the salivary glands synthesize high levels of proteins, such as glue protein to form a sticky matrix that allows the larva to adhere itself to solid surfaces to prepare for pupation. For several of the tagged RPs a clear signal was also apparent in the DAPI-stained region of the nucleus. While this nuclear signal was more apparent with some of the proteins (S5a, S11, S18, L8 and L11), it was also visible to a smaller extent with the other proteins. In summary, the nucleolar/cytoplasmic sub-cellular localization of the tagged protein is consistent with them being incorporated into ribosomes, and the accumulation in the DAPI-stained region of the nucleus suggests that the proteins can associate with chromosomes.

Figure 4 Sub-cellular localization of GFP tagged 40S RPs in salivary glands.

Leftmost panels show micrographs of salivary gland cells expressing the respective GFP tagged 40S RPs. Middle panels show DAPI staining of the nucleus. Right panels show merged images of the GFP and DAPI signal. The images are confocal microscopy sections taken with a 40X oil immersion objective. Both wide (A) and close-up (B) views are shown.

Figure 5 Sub-cellular localization of RFP tagged 60S RPs in salivary glands.

Leftmost panels show micrographs of salivary gland cells expressing the respective RFP tagged 60S RPs. Middle panels show DAPI staining of the nucleus. Right panels show merged images of the RFP and DAPI signal. The images are confocal microscopy sections taken with a 40X oil immersion objective. Both wide (A) and close-up (B) views are shown.

Tagged RPs associate with transcribed regions on the chromosomes

The presence of the tagged proteins in the DAPI-stained region of the nucleus suggested association with chromosomes. To investigate this further, we used the polytene chromosome system. The Drosophila polytene chromosomes from third instar larval salivary glands are a powerful tool to investigate the association of proteins with chromatin and allow discrimination between transcribed and non-transcribed regions. On polytene chromosomes, the transcribed regions (interbands) are cytologically distinct from regions which are not transcribed (bands) (Zhimulev et al., 2004). Transgenic flies expressing three representatives of the tagged proteins were used: UAS-RpS9-GFP, UAS-RpS18-GFP and UAS-RpL11-RFP. The transgenes were expressed by crossing them with the SG-Gal4 driver described above. The larval progeny were examined under a UV dissecting microscope and salivary glands were dissected from larvae that appeared fluorescent under the microscope. Polytene chromosome squashes were prepared and the tagged RPs were visualized with either anti-GFP or anti-RFP antibodies (Material and Methods). The signal was visible most apparently at interbands (Fig. 6A, darker region, poorly stained by DAPI). In some nuclei it was apparent that RPs are highly concentrated at heat shock puffs (indicated by the arrows in the bottom panels of Fig. 6A). These puffs are highly transcribed regions that encode heat shock proteins and appear as areas of loosely packed chromatin (Zhimulev et al., 2004); they are often induced without deliberate heat shock during dissection. The strong staining at the chromosome puff indicates that the RPs association strongly correlates to transcription activity. No signal was apparent on chromosomes prepared from the parental y w strain and stained with either the GFP or RFP antibody (Fig. 6B).

Figure 6 Tagged RPs associate with specific chromosomal regions.

(A) Leftmost panels show polytene chromosome immunostaining with either an anti-GFP primary antibody (RpS9-GFP and RpS18-GFP) or an anti-RFP (RpL11-RFP), both detected with a Cy3-conjugated (red signal) secondary antibody. Middle panels show DAPI staining of the chromosomes. Rightmost panels are merged images of Cy3 and DAPI signals. Note that Cy3 and DAPI show complementary pattern: the Cy3 signal is most intense at interbands, while DAPI mostly stains bands. Bottom rows show magnified view of two heat-shock transcription loci enriched with RpL11-RFP (RpL11-RFP, arrows). (B) Top row shows immunostaining of chromosome squashes from the parental yw strain not carrying the GFP tagged transgenes, with the same anti-GFP used for Fig. 4. Bottom row shows immunostaining of the parental strain with the anti-RFP used in Fig. 5.

Chromosome association of genomically tagged RpL41

Although the data described above indicated that tagged RPs can associate with chromosomes, it is possible that the association might have been affected by the fact that the transgenes were inserted at heterologous positions and were expressed with the UAS/GAL4, which probably produced different levels of expression relative to the endogenous RP promoters. More importantly, the UAS transgenes were expressed in cells that already expressed the corresponding endogenous genes and therefore, the tagged proteins are expected to compete with the endogenous proteins for incorporation into ribosomal subunits. To address this concern, we have analyzed the sub-cellular and chromosomal association of RpL41, which had been previously tagged at the natural genomic locus by inserting a YFP exon into the first intron of the endogenous gene (Fig. 7A). RpL41 is a small protein which is not associated with the isolated 40S or 60S subunit but specifically with the 80S; RpL41 is positioned at the intersubunit surface (Klinge et al., 2012). YFP-RpL41 appears to be functional because the tagged gene is homozygous viable; the RpL41 gene, similarly to other RP genes is found as a single copy in the genome of Drosophila and is expected to be essential for viability (Marygold et al., 2007). Microscopy imaging of the glands dissected from homozygous flies showed that RpL41 has a sub-cellular localization similar to that of the tagged RPs described above: the protein is abundant in both the cytoplasm and the nucleus with the characteristic nucleolar enrichment (Fig. 7B). Notably the signal in the DAPI-stained region of the nucleus was as strong as that seen with RpS18-GFP and RpL11-RFP, which were the tagged RPs with the most prominent nuclear staining. Analysis of chromosomes squashes confirmed its association with chromosomes and indicated that the RpL41 is most abundant at transcribed regions (Fig. 7C; arrows in the insets on the right indicate two prominent interbands). In this instance chromosomes were squashed in 50% glycerol instead of the standard acetic acid-containing solution which destroys the protein fluorescence. Standard squashes, followed by immunostaining show a reduced chromosomal signal (not shown) probably because RpL41, being only 25 amino acids long, is poorly fixed by the standard fixation procedure and readily disassociates during the immunostaining. Therefore, this characterization clearly shows that this RP associates with chromosomal loci.

Figure 7 Endogenous expression of YFP tagged RpL41 under native promoter control.

(A) Schematic of the genomic tagging of RpL41 at its endogenous locus with YFP. (B) Confocal section showing YFP fluorescence in salivary gland cells expressing YFP-RpL41. Images on the right show a magnified view of a cell; middle panel shows DAPI staining and that at the bottom shows a merge of the YFP and DAPI signals. Arrows indicate the nucleolus. (C) Polytene chromosome spread in native conditions. Top-left diagram shows YFP fluorescence corresponding to YFP-RpL41, and bottom-left panel shows a merged image of the YFP fluorescence with DAPI staining. Arrows indicate a fragment of the nucleolus. The rightmost insets are magnified images of the area boxed in the main picture, showing YFP signal (upper) and DAPI staining (lower). Lines indicate intense fluorescence at two prominent interbands. The images were visualised under a standard epi-fluorescence microscope.

Discussion

The characterization of the transgenes we have described here indicates that the RPs we have tagged can associate with chromosomes. While it is feasible that this association might be affected by the GFP or RFP tag, a similar association is apparent with all the RPs that were analyzed regardless of whether the tag was GFP or RFP. Although the tagged RPs detected in the nucleus might correspond to free proteins not incorporated into ribosomal subunits, the characteristic nucleolar enrichment we have observed and their presence in polysomal fractions also suggest that these tagged RPs can be incorporated into ribosomes as previously reported for some of these proteins in other organisms (Hurt et al., 1999; Inada et al., 2002; Lam et al., 2007). Notably, similarly tagged RpS18 and RpL11 can genetically complement deletions in the corresponding genes, further arguing that such tagged RPs can be incorporated into functional ribosomal subunits (data not shown). These observations are consistent with the earlier study which indicated the presence of RpS15, RpL32 and other RPs at transcription sites (Brogna, Sato & Rosbash, 2002; De & Brogna, 2010; De et al., 2011). Whereas the earlier study used antibodies directed against endogenous RPs, and it could be argued that the antibodies were detecting cross-reacting antigens, here the protein accumulation at the chromosomes was directly visualized by GFP or RFP fluorescence in intact cells; and, the chromosomal immunostaining was with either GFP or RFP antibodies which did not show a noticeable cross-reactivity to endogenous chromosomal proteins (Fig. 6B). Remarkably, chromosomal association was also apparent for RpL41 which was functionally tagged with YFP at its genomic locus. Although the data we present do not allow the resolution of the issue of whether RPs detected at the sites of transcription are actually part of complete ribosome subunits, they clearly confirm their presence at these sites; and the fact that several RPs are present suggests that they might be recruited as complexes, possibly ribosomal subunits, as previously proposed (Brogna, Sato & Rosbash, 2002; De & Brogna, 2010). As reviewed in the introduction, the current understanding is that many RPs with non-ribosomal function exert these functions while they are off the subunits; however, in view of the data we show here and of those recently reported by De et al. (2011) using fission yeast, it is possible that these RPs are in ribosomal subunits or other ribosomal-like complexes.

Finally, it has recently been proposed that RP composition might vary between functional ribosomes (Kondrashov et al., 2011; Lee, Burdeinick-Kerr & Whelan, 2013). It is thus feasible that the presence or absence of some of the RPs might be a characteristic of nuclear ribosomes; this model would for example provide an explanation for why some of the tagged RPs seem more abundant than others at the chromosomes in our transgenic flies.

Material and Methods

Plasmids construction and transgenes

Unless indicated otherwise, cloning was done using the Gateway technology (Invitrogen). The entire coding regions of RPs were PCR amplified from cDNA libraries which were provided by the Drosophila Genomics Resource Centre (DGRC) (www.dgrc.cgb.indiana.edu). Entry clones were generated in the pDONR-221 vector and then the DNA inserts were recombined into either the destination vector pAGW (N-terminal GFP fusions) or into pAWG (GFP C-terminal fusions), both of which were obtained from DGRC; these plasmids are part of the Drosophila Gateway Vector Collection produced by Dr Terence Murphy (www.ciwemb.edu/labs/murphy/Gateway%20vectors.html). To generate the transgenes, RpS9, RpS18 and RpL11 were cloned into either pTWG (C-terminal GFP fusions) or pTWR (C-terminal RFP fusions) (DGRC). The other transgenes (RpS2-GFP, RpS5a, RpS11, RpS13, RpL8, RpL32 and RpL36) were generated using the PhiC31 integrase-mediated transgenesis system which is based on the site-specific bacteriophage PhiC31 integrase (Bischof et al., 2007). To generate pUAST derivatives compatible with the PhiC31 transformation system, a full length Gateway recombination cassette from pTWG and pTWR was subcloned into the PhiC31-compatible pUAST derivative (pUASTattB). The Gateway recombination cassette, including the GFP (attR1-Cmr-ccdB-attR2-GFP) and RFP (attR1-Cmr-ccdB-attR2-RFP) regions, were PCR amplified from pTWG and pTWR with primers KJ67 and KJ68 (listed below). Both primers carry KpnI restriction sites at the 5’ end. The PCR products were digested and inserted into the KpnI site in the multiple cloning site of pUASTattB (Bischof et al., 2007); this generated pUAST.attB.WG and pUAST.attB.WR. RP coding regions were PCR amplified as above and cloned into either plasmid. 40S RP constructs were cloned in pUAST.attB.WG (GFP fusions) and 60S RP into pUAST.attB.WR (RFP fusions). The primers used to PCR the RP cDNA are listed below. All transgenes were generated by Bestgene (USA) using either P element mediated transformation of a standard yw strain or injecting strains carrying specific PhiC31 recombination sites: Strain 24484 (position 58A) to insert RpS13, RpS11, RpS2 and RpS5a; and Strain 24482 (position 51C) to insert RpL36, RpL32, RpL23 and RpL8. The YFP-RpL41 (CPTI-002881) strain was produced by Flyprot (http://www.flyprot.org/). SG-Gal4 is a heat-shock-Gal4 transgene available in the laboratory which, as other similar constructs, is characteristically expressed in the salivary glands (without heat shocking) throughout the 3rd instar larval stage (Gerlitz et al., 2002); this was considered preferable to late-expressing drivers such as glue-proteins drivers to maximize incorporation into ribosomal subunits.

Primers:

RpS2 Forward (KJ64): GGGGACAAGTTTGTACAAAAAAGCAGGCTTCACCATGGCGGACGAAGCTCCAGCC

RpS2 Reverse (KJ66): GGGGACCACTTTGTACAAGAAAGCTGGGTCGGCATCGGCGTGCAGACG

RpS5a Forward (KJ51): GGGGACAAGTTTGTACAAAAAAGCAGGCTTCACCATGGCCGAAGTTGCTGAAAAC

RpS5a Reverse (KJ53): GGGGACCACTTTGTACAAGAAAGCTGGGTCACGGTTGGACTTGGCGAC

RpS9 Forward (KJ4): GGGGACAAGTTTGTACAAAAAAGCAGGCTTCACCATGGTGAACGGCCGCATACC

RpS9 Reverse (KJ5): GGGGACCACTTTGTACAAGAAAGCTGGGTCGTCCTCCTCCTCTTCAGCA

RpS11 Forward (KJ58): GGGGACAAGTTTGTACAAAAAAGCAGGCTTCACCATGGCTGATCAGAACGAGCGC

RpS11 Reverse (KJ60): GGGGACCACTTTGTACAAGAAAGCTGGGTCGTACTTCTTGAAGCTCTT

RpS13 Forward (KJ54): GGGGACAAGTTTGTACAAAAAAGCAGGCTTCACCATGGGTCGTATGCACGCTCCT

RpS13 Reverse (KJ57): GGGGACCACTTTGTACAAGAAAGCTGGGTCGGCAACC

RpS15 Forward (KJ8): GGGGACAAGTTTGTACAAAAAAGCAGGCTTCACCATGGCCGATCAAGTCGATGAAAA

RpS15 Reverse (KJ9): GGGGACAAGTTTGTACAAGAAAGCTGGGTCCTTCAGAGGAATGAAACG

RpS18 Forward (KJ13): GGGGACAAGTTTGTACAAAAAAGCAGGCTTCACCATGTCGCTCGCTCGTCATCCCAGAGA

RpS18 Reverse (KJ14): GGGGACAAGTTTGTACAAGAAAGCTGGGTCCTTCTTCTTGGACACACCCAC

RpL8 Forward (KJ39): GGGGACAAGTTTGTACAAAAAAGCAGGCTTCACCATGGGTCGCGTTATTCGTGCA

RpL8 Reverse (KJ41): GGGGACCACTTTGTACAAGAAAGCTGGGTCCTTGTCCTTGCTGTCGCC

RpL11 Forward (KJ22): GGGGACAAGTTTGTACAAAAAAGCAGGCTTCACCATGGCGGTAGGTTCAACCAC

RpL11 Reverse (KJ21): GGGGACCACTTTGTACAAGAAAGCTGGGTCCTTCTTGGTGTTCAAGATG

RpL32 Forward (KJ42): GGGGACAAGTTTGTACAAAAAAGCAGGCTTCACCATGACCATCCGCCCAGCATACA

RpL32 Reverse (KJ44): GGGGACCACTTTGTACAAGAAAGCTGGGTCCTCGTTCTCTTGAGAACG

RpL36 Forward (KJ36): GGGGACAAGTTTGTACAAAAAAGCAGGCTTCACCATGGCAGTGCGCTACGAGCT

RpL36 Reverse (KJ38): GGGGACCACTTTGTACAAGAAAGCTGGGTCCTTGGCGTGGGTCTGGGC

pTW Forward (KJ67): GGGGGTACCGAGAACTCTGAATAGGGAATTG

pTW Reverse (KJ68): GGGGGTACCAGATCCTCTAGCTTACGTCA

Cell culture, transfection and microscopy

D. melanogaster Schneider line-2 cells (S2 cells) were grown in Insect-XPRESS medium (Cambrex) with 4% fetal bovine serum, 1% penicillin/streptomycin/glutamine mix (Cambrex), and grown at 27 °C without CO2 . Transfection was typically done in 6-well plates, seeded the night before with 3 × 106 cells/well. Transfection was performed using dimethyl dioctadecyl ammonium bromide (DDAB, Sigma) using modifications of a previously described protocol (Ramanathan et al., 2008). Plasmid DNA was diluted in serum-free media (3.75 µg typically), mixed with DDAB (typically 45 µL of a 400 µg/mL DDAB stock), incubated 30 min at room temperature and then added to the cells, which had previously been washed twice with serum-free media and kept in 0.875 mL of the serum-free media. The transfection mix was incubated at 27 °C for 5 h. After 5 h, the media was replaced by 2 mL of complete media containing serum and antibiotics. Cells were incubated one or two nights at 27 °C prior fixation with 4% formaldehyde in PBS, pH 7.4, for 15 min at 20 °C. The cells were washed in PBS, pH 7.4, three times, 10 min each and permeabilized in 0.05% Tween 20 in PBS for 5 min on ice, and finally washed again in PBS three times, 10 min each. DAPI (4’-6-diamidino-2-phenylinodole) (Sigma-Aldrich) was added to the second wash (0.1 µg/mL) to stain the DNA. The coverslip was mounted with a drop of mounting medium (PromoFluor Antifade Reagent, PromoKine) and was sealed with clear nail polish to prevent drying and movement under the microscope. Microscopy imaging was carried out with either a Leica DMIRE2 or a Nikon Eclipse Ti epifluorescence microscope, equipped with CCD cameras (ORCA, Hamamatsu Photonics). Confocal images were acquired with a Leica SP2-AOBS microscope. The Open Lab software (Improvision), or the Nikon NIS or the Leica confocal software were used to acquire the images which were subsequently processed using the ImageJ software (rsbweb.nih.gov/ij/).

Polysome analysis

Transfected S2 cells (typically one day after transfection) were briefly (15 min) treated with 100 µg/mL cycloheximide and centrifuged at 4 °C. The pellet was washed in cold PBS and then lysed in 600 µL lysis buffer containing 20 mM HEPES, KOH pH 7.4, 2 mM magnesium acetate, 100 mM Potassium acetate, 1 mM dithiothreitol (DTT), 250 µg/mL heparin, 0.05 mM aurintricarboxylic acid (ATA, Sigma), 0.25% Triton X-100, 100 µg/mL cycloheximide and EDTA-free protease inhibitor cocktail (Roche). The lysate was cleared by centrifugation at 13,000 rpm for 20 min and the A260 of the extracts was measured. Then 12–20 A260 units of lysate were centrifuged through a 10%–50% sucrose gradient at 38,000 rpm for 3 h in a Beckman SW40Ti rotor. All of the above steps were done at 4 °C. After centrifugation, the gradients were pumped (from the bottom, using a steel capillary) through a flow-through UV spectrophotometer (Pharmacia LKB-Optical Unit UV-1) with a peristaltic pump (P-1, Pharmacia) at a speed of 1.2 mL/min. The A254 was recorded as the fractions passed through the flow cell. After fractionation, the proteins were precipitated using the trichloroacetic acid (TCA) and Na-deoxycholate (DOC) method as described in with some modifications as further described. To 1 mL of fractions, 10 µL of 1.25% Na-deoxycholate (DOC) was added to a final concentration of 125 µg/mL, the mixture was vortexed, and allowed to sit at room temperature for 15 min. 350 µL of 24% TCA was added, the mixture vortexed and centrifuged at 4 °C at maximum speed for 30 min. The supernatant was carefully decanted and the precipitate was washed with ice cold acetone by spinning for 2 min at 4 °C. The precipitate was then re-suspended in 40 µL of 2x SDS gel loading buffer with 5% β-mercaptoethanol and the protein was denatured by boiling for 5 min. The protein extract was kept on ice for 2 min and centrifuged at 4 °C for 5 min before loading.

Salivary gland dissection and native or standard polytene chromosome immunostaining

Third-instar wandering larvae were dissected in PBS (removal of most of the fat body attached to the glands was required to obtain a clear chromosome spread). For imaging of the intact cells, glands were fixed in cold 4% formaldehyde PBS for 30 min and processed as described above for cells. To analyse protein association with the polytene chromosomes we modified two published protocols: using either acetic acid (Shopland & Lis, 1996); or glycerol treatment to facilitate chromosome spreading (Johansen et al., 2009). In one protocol fixed glands were incubated for 30 min in 50% glycerol containing DAPI (0.1 µg/mL) at room temperature, followed by squashing in a 20–30 µL drop of the same glycerol/DAPI solution on a microscope slide. Unlike the standard protocol this procedure does not require treatment with acetic acid, which inhibits fluorescence from GFP or similar proteins. Squashing consisted in covering the drop with a coverslip and tapping on it with the blunt end of forceps (or a similarly shaped object) to break the nuclei and spread the chromosomes. Because the glands are not treated with acetic acid chromosome spreading is not as effective as with the standard procedure (described below). The slide/coverslip was placed between folded paper tissue and the glands squashed on the microscope slide by forcing on the coverslip with the thumb on a flat surface. The quality of the squashes was routinely checked with phase contrast microscope and good squashes were imaged by epifluorescence microscopy as described below. The other polytene squashing protocol is a standard procedure that involves fixation in acetic acid and the protocol used here is actually a slight modification of what has been previously described (Shopland & Lis, 1996). Glands were dissected in solution A (15 mM HEPES pH 7.4, 60 mM KCl, 15 mM NaCl, 1.5 mM Spermine, 1.5 mM Spermidine) plus 10% Triton X-100 (this is required to produce a clean chromosome spread but can be omitted). Straight after dissection the glands were fixed for about one minute in solution A supplemented with 4% paraformaldehyde (EMS). The glands were further fixed in 50% acetic acid and 4% paraformaldehyde (EM grade, EMS) for 3–5 min and squashed as described above. Good squashes were frozen in liquid nitrogen and coverslips were removed with a razor blade. The position of the coverslip was marked on the slide with a diamond pencil, then the slide was submerged in 95% ethanol in a Coplin jar (or similar) and stored at −20 °C (the slide can be processed straight away for immunostaining or kept for 1–2 days in the freezer). Prior to immunostaining, the slides were rehydrated by immerging them in 50% ethanol and 50% TBS solution for 10 min, and then rinsed twice with TBS (150 mM NaCl, 10 mM Tris-Cl pH 7.0–7.5, 0.05% Tween). Rehydrated slides were blocked in blocking solution containing TBS, 10% Fetal Bovine Serum (FBS) and 0.05% sodium azide (NaN3) for 50–60 min at room temperature. 20 µL of diluted primary antibody (1:100) in 4% blocking solution was put on a clean coverslip on the bench as a droplet. The tagged RPs were detected using either an antibody specific to EGFP (Anti-GFP rabbit IgG (Molecular Probes, Invitrogen)) or an antibody specific to mRFP (rabbit IgG anti-RFP (Millipore)). Tissue dried (outside the chromosome area) slides were lowered onto the coverslip, and care was taken to pick it up in the chromosome region of slide, that was previously marked by the diamond pencil. Slides were then incubated in a humid chamber that contained TBS, at room temperature for 1–2 h. After incubation, coverslips were removed by tapping the slide on the side of a beaker and slides were rinsed three times in TBS for 10 min each time at room temperature. The secondary antibody procedure was similar to the primary antibody staining; the antibody was diluted in 4% blocking solution (1:400). The secondary antibodies used were either fluorescein isothiocyanate (FITC) conjugated with goat anti mouse IgM or Cyanine 3 (Cy3) conjugated with donkey anti-rabbit IgG. All secondary antibodies were purchased from Jackson Immuno Research Technologies. Slides were incubated with the secondary antibody at room temperature for 1–2 h and then washed three times in PBS. DAPI was added to TBS at a 1:10,000 dilution (0.1 µg/mL) to the second wash. Slides were air dried and mounted as described above for cells. Stained polytene chromosomes were inspected by epi-fluorescence microscopy with a 40X dry objective lens. Microscopy was as described for cells above.

Additional Information and Declarations

Competing Interests

Author Contributions

Saverio Brogna is an Academic Editor for PeerJ. The authors have no other competing interests.

Kushal Nivriti Rugjee analyzed the data, wrote the paper.

Subhendu Roy Chaudhury performed the experiments, analyzed the data.

Khalid Al-Jubran conceived and designed the experiments, performed the experiments, analyzed the data.

Preethi Ramanathan and Tina Matina performed the experiments.

Jikai Wen analyzed the data, contributed reagents/materials/analysis tools.

Saverio Brogna conceived and designed the experiments, analyzed the data, contributed reagents/materials/analysis tools, wrote the paper.

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
