# Peer review of "Fluorescent protein tagging confirms the presence of ribosomal proteins at Drosophila polytene chromosomes"

_PeerJ, doi:10.7717/peerj.15_

## Round 0.1 · original submission · Minor Revisions

Both reviewers find your manuscript interesting and suitable for the PeerJ journal, however, they suggest several changes that may improve your manuscript. I would like to ask you to revise your manuscript accordingly and resubmit.

·

Basic reporting

1. Correct miss-localization (page 4).

2. Correct coversplip (page 13)

3. Reference Format is not correct.

Experimental design

1. What do you mean with this sentence? The nuclear localization of the tagged proteins suggested association with chromosomes.

2. How many cells were analyzed for each variant of transfection with fluorescent/confocal microscope?

3. I miss scale bar for all microscopical images. What is it low and high magnification in Figure 4 & 5?

4. What kind of zoom factor was used in “low magnification”?

5. In Figure 2 D, S15-GFP image does not totally correspond to merge image.

6. Page 12: „Cells were incubated one ore two nights..“ Do you mean that cells were cultivated overnight and 30 hours?

Validity of the findings

1. Could you please suggest some future use of your results?

·

Basic reporting

No Comments

Experimental design

No comments

Validity of the findings

No Comments

Additional comments

This is a careful study of the in vivo localisation of selected GFP-tagged ribosomal proteins in Drosophila salivary glands. The study supports the hypothesis that at least some ribosomal proteins are associated with chromatin in vivo. While not essential for publication, it would be good to know if the GFP-tagged RPs can rescue mutations in the endogenous genes. Similarly it would be interesting to examine the degree of co-localisation seen with different RPs, presumably why RFP and GFP versions were tagged.

---

## Round 0.2 · accepted · Accept

Thank you for supporting PeerJ and Open Publishing.